# PRIVATE POST-GAN BOOSTING

**Marcel Neunhoeffer**
University of Mannheim
mneunhoe@mail.uni-mannheim.de

**Zhiwei Steven Wu**
Carnegie Mellon University
zstevenwu@cmu.edu

**Cynthia Dwork**
Harvard University
dwork@seas.harvard.edu

## ABSTRACT

Differentially private GANs have proven to be a promising approach for generating realistic synthetic data without compromising the privacy of individuals. Due to the privacy-protective noise introduced in the training, the convergence of GANs becomes even more elusive, which often leads to poor utility in the output generator at the end of training. We propose *Private post-GAN boosting (Private PGB)*, a differentially private method that combines samples produced by the sequence of generators obtained during GAN training to create a high-quality synthetic dataset. To that end, our method leverages the Private Multiplicative Weights method (Hardt and Rothblum, 2010) to reweight generated samples. We evaluate Private PGB on two dimensional toy data, MNIST images, US Census data and a standard machine learning prediction task. Our experiments show that Private PGB improves upon a standard private GAN approach across a collection of quality measures. We also provide a non-private variant of PGB that improves the data quality of standard GAN training.

## 1 INTRODUCTION

The vast collection of detailed personal data, including everything from medical history to voting records, to GPS traces, to online behavior, promises to enable researchers from many disciplines to conduct insightful data analyses. However, many of these datasets contain sensitive personal information, and there is a growing tension between data analyses and data privacy. To protect the privacy of individual citizens, many organizations, including Google (Erlingsson et al., 2014), Microsoft (Ding et al., 2017), Apple (Differential Privacy Team, Apple, 2017), and more recently the 2020 US Census (Abowd, 2018), have adopted *differential privacy* (Dwork et al., 2006) as a mathematically rigorous privacy measure. However, working with noisy statistics released under differential privacy requires training.

A natural and promising approach to tackle this challenge is to release *differentially private synthetic data*—a privatized version of the dataset that consists of fake data records and that approximates the real dataset on important statistical properties of interest. Since they already satisfy differential privacy, synthetic data enable researchers to interact with the data freely and to perform the same analyses even without expertise in differential privacy. A recent line of work (Beaulieu-Jones et al., 2019; Xie et al., 2018; Yoon et al., 2019) studies how one can generate synthetic data by incorporating differential privacy into *generative adversarial networks* (GANs) (Goodfellow et al., 2014). Although GANs provide a powerful framework for synthetic data, they are also notoriously hard to train and privacy constraint imposes even more difficulty. Due to the added noise in the private gradient updates, it is often difficult to reach convergence with private training.

In this paper, we study how to improve the quality of the synthetic data produced by private GANs. Unlike much of the prior work that focuses on fine-tuning of network architectures and training techniques, we propose *Private post-GAN boosting* (Private PGB)—a differentially private method that boosts the quality of the generated samples after the training of a GAN. Our method can be viewed as a simple and practical amplification scheme that improves the distribution from any ex-

isting black-box GAN training method – private or not. We take inspiration from an empirical observation in Beaulieu-Jones et al. (2019) that even though the generator distribution at the end of the private training may be a poor approximation to the data distribution (due to e.g. mode collapse), there may exist a high-quality mixture distribution that is given by several generators over different training epochs. PGB is a principled method for finding such a mixture at a moderate privacy cost and without any modification of the GAN training procedure.

To derive PGB, we first formulate a two-player zero-sum game, called *post-GAN* zero-sum game, between a *synthetic data* player, who chooses a distribution over generated samples over training epochs to emulate the real dataset, and a *distinguisher* player, who tries to distinguish generated samples from real samples with the set of discriminators over training epochs. We show that under a "support coverage" assumption the synthetic data player's mixed strategy (given by a distribution over the generated samples) at an equilibrium can successfully "fool" the distinguisher–that is, no mixture of discriminators can distinguish the real versus fake examples better than random guessing. While the strict assumption does not always hold in practice, we demonstrate empirically that the synthetic data player's equilibrium mixture consistently improves the GAN distribution.

The Private PGB method then privately computes an approximate equilibrium in the game. The algorithm can be viewed as a computationally efficient variant of MWEM (Hardt & Rothblum, 2010; Hardt et al., 2012), which is an inefficient query release algorithm with near-optimal sample complexity. Since MWEM maintains a distribution over exponentially many "experts" (the set of all possible records in the data domain), it runs in time exponential in the dimension of the data. In contrast, we rely on private GAN to reduce the support to only contain the set of privately generated samples, which makes PGB tractable even for high-dimensional data.

We also provide an extension of the PGB method by incorporating the technique of *discriminator rejection sampling* (Azadi et al., 2019; Turner et al., 2019). We leverage the fact that the distinguisher's equilibrium strategy, which is a mixture of discriminators, can often accurately predict which samples are unlikely and thus can be used as a rejection sampler. This allows us to further improve the PGB distribution with rejection sampling without any additional privacy cost since differential privacy is preserved under post-processing. Our Private PGB method also has a natural non-private variant, which we show improves the GAN training without privacy constraints.

We empirically evaluate both the Private and Non-Private PGB methods on several tasks. To visualize the effects of our methods, we first evaluate our methods on a two-dimensional toy dataset with samples drawn from a mixture of 25 Gaussian distributions. We define a relevant quality score function and show that the both Private and Non-Private PGB methods improve the score of the samples generated from GAN. We then show that the Non-Private PGB method can also be used to improve the quality of images generated by GANs using the MNIST dataset. Finally, we focus on applications with high relevance for privacy-protection. First we synthesize US Census datasets and demonstrate that the PGB method can improve the generator distribution on several statistical measures, including 3-way marginal distributions and pMSE. Secondly, we evaluate the PGB methods on a dataset with a natural classification task. We train predictive models on samples from Private PGB and samples from a private GAN (without PGB), and show that PGB consistently improves the model accuracy on real out-of-sample test data.

**Related work.** Our PGB method can be viewed as a modular boosting method that can improve on a growing line of work on differentially private GANs (Beaulieu-Jones et al., 2019; Xie et al., 2018; Frigerio et al., 2019; Torkzadehmahani et al., 2020). To obtain formal privacy guarantees, these algorithms optimize the discriminators in GAN under differential privacy, by using private SGD, RMSprop, or Adam methods, and track the privacy cost using moments accounting Abadi et al. (2016); Mironov (2017). Yoon et al. (2019) give a private GAN training method by adapting ideas from the PATE framework (Papernot et al., 2018).

Our PGB method is inspired by the Private Multiplicative Weigths method (Hardt & Rothblum, 2010) and its more practical variant MWEM (Hardt et al., 2012), which answer a large collection of statistical queries by releasing a synthetic dataset. Our work also draws upon two recent techniques (Turner et al. (2019) and Azadi et al. (2019)) that use the discriminator as a rejection sampler to improve the generator distribution. We apply their technique by using the mixture discriminator computed in PGB as the rejection sampler. There has also been work that applies the idea of boosting to (non-private) GANs. For example, Arora et al. (2017) and Hoang et al. (2018) propose methods

that directly train a mixture of generators and discriminators, and Tolstikhin et al. (2017) proposes AdaGAN that reweighes the real examples during training similarly to what is done in AdaBoost (Freund & Schapire, 1997). Both of these methods may be hard to make differentially private: they either require substantially more privacy budget to train a collection of discriminators or increase the weights on a subset of examples, which requires more adding more noise when computing private gradients. In contrast, our PGB method boosts the generated samples *post* training and does not make modifications to the GAN training procedure.

## 2 PRELIMINARIES

Let $\mathcal{X}$ denote the data domain of all possible observations in a given context. Let $p_d$ be a distribution over $\mathcal{X}$. We say that two datasets $X, X' \in \mathcal{X}^n$ are adjacent, denoted by $X \sim X'$, if they differ by at most one observation. We will write $p_X$ to denote the empirical distribution over $X$.

**Definition 1** (Differential Privacy (DP) (Dwork et al., 2006)). *A randomized algorithm $\mathcal{A} : \mathcal{X}^n \to \mathcal{R}$ with output domain $\mathcal{R}$ (e.g. all generative models) is $(\varepsilon, \delta)$-differentially private (DP) if for all adjacent datasets $X, X' \in \mathcal{X}^n$ and for all $S \subseteq \mathcal{R}$: $P(\mathcal{A}(X) \in S) \leq e^\varepsilon P(\mathcal{A}(X') \in S) + \delta$.*

A very nice property of differential privacy is that it is preserved under post-processing.

**Lemma 1** (Post-processing). *Let $\mathcal{M}$ be an $(\varepsilon, \delta)$-differentially private algorithm with output range $R$ and $f : R \to R'$ be any mapping, the composition $f \circ \mathcal{M}$ is $(\varepsilon, \delta)$-differentially private.*

As a result, any subsequent analyses conducted on DP synthetic data also satisfy DP.

The *exponential mechanism* (McSherry & Talwar, 2007) is a private mechanism for selecting among the best of a discrete set of alternatives $\mathcal{R}$, where "best" is defined by a quality function $q \colon \mathcal{X}^n \times \mathcal{R} \to \mathcal{R}$ that measures the quality of the result $r$ for the dataset $X$. The sensitivity of the quality score $q$ is defined as $\Delta(q) = \max_{r \in \mathcal{R}} \max_{X \sim X'} |q(X, r) - q(X', r)|$. Then given a quality score $q$ and privacy parameter $\varepsilon$, the exponential mechanism $\mathcal{M}_E(q, \varepsilon, X)$ simply samples a random alternative from the range $\mathcal{R}$ such that the probability of selecting each $r$ is proportional to $\exp(\varepsilon q(X, r) / (2\Delta(q)))$.

### 2.1 DIFFERENTIALLY PRIVATE GAN

The framework of *generative adversarial networks (GANs)* (Goodfellow et al., 2014) consists of two types of neural networks: *generators* and *discriminators*. A generator $G$ is a function that maps random vectors $z \in Z$ drawn from a prior distribution $p_z$ to a sample $G(z) \in \mathcal{X}$. A discriminator $D$ takes an observation $x \in \mathcal{X}$ as input and computes a probability $D(x)$ that the observation is real. Each observation is either drawn from the underlying distribution $p_d$ or the induced distribution $p_g$ from a generator. The training of GAN involves solving the following joint optimization over the discriminator and generator:

$$\min_G \max_D \ \mathbb{E}_{x \sim p_X}[f(D(x))] + \mathbb{E}_{z \sim p_z}[f(1 - D(G(z)))]$$

where $f \colon [0, 1] \to \mathbb{R}$ is a monotone function. For example, in standard GAN, $f(a) = \log a$, and in Wasserstein GAN (Arjovsky et al., 2017), $f(a) = a$. The standard (non-private) algorithm iterates between optimizing the parameters of the discriminator and the generator based on the loss functions:

$$L_D = -\mathbb{E}_{x \sim p_X}[f(D(x))] - \mathbb{E}_{z \sim p_z}[f(1 - D(G(z)))], \quad L_G = \mathbb{E}_{z \sim p_z}[f(1 - D(G(z)))]$$

The private algorithm for training GAN also performs the same alternating optimization, but it optimizes the discriminator under differential privacy while keeping the generator optimization the same. In general, the training proceeds over epochs $\tau = 1, \ldots, N$, and at the end of each epoch $\tau$ the algorithm obtains a discriminator $D_\tau$ and a generator $G_\tau$ by optimizing the loss functions respectively. In Beaulieu-Jones et al. (2019); Xie et al. (2018), the private optimization on the discriminators is done by running the private SGD method Abadi et al. (2016) or its variants. Yoon et al. (2019) performs the private optimization by incorporating the PATE framework Papernot et al. (2018). For all of these private GAN methods, the entire sequence of discriminators $\{D_1, \ldots, D_N\}$ satisfies privacy, and thus the sequence of generators $\{G_1, \ldots, G_N\}$ is also private since they can be viewed as post-processing of the discriminators. Our PGB method is agnostic to the exact private GAN training methods.

## 3 PRIVATE POST-GAN BOOSTING

The noisy gradient updates impede convergence of the differentially private GAN training algorithm, and the generator obtained in the final epoch of the training procedure may not yield a good approximation to the data distribution. Nonetheless, empirical evidence has shown that a mixture over the set of generators can be a realistic distribution (Beaulieu-Jones et al., 2019). We now provide a principled and practical scheme for computing such a mixture subject to a moderate privacy budget. Recall that during private GAN training method produces a sequence of generators $\mathcal{G} = \{G_1, \ldots, G_N\}$ and discriminators $\mathcal{D} = \{D_1, \ldots, D_N\}$. Our boosting method computes a weighted mixture of the $G_j$'s and a weighted mixture of the $D_j$'s that improve upon any individual generator and discriminator. We do that by computing an equilibrium of the following *post-GAN (training)* zero-sum game.

### 3.1 POST-GAN ZERO-SUM GAME.

We will first draw $r$ independent samples from each generator $G_j$, and let $B$ be the collection of the $rN$ examples drawn from the set of generators. Consider the following *post-GAN* zero-sum game between a *synthetic data player*, who maintains a distribution $\phi$ over the data in $B$ to imitate the true data distribution $p_X$, and a *distinguisher player*, who uses a mixture of discriminators to tell the two distributions $\phi$ and $p_X$ apart. This zero-sum game is aligned with the minimax game in the original GAN formulation, but is much more tractable since each player has a finite set of strategies. To define the payoff in the game, we will adapt from the Wasserstein GAN objective since it is less sensitive than the standard GAN objective to the change of any single observation (changing any single real example changes the payoff by at most $1/n$), rendering it more compatible with privacy tools. Formally, for any $x \in B$ and any discriminator $D_j$, define the payoff as

$$U(x, D_j) = \mathbb{E}_{x' \sim p_X} [D_j(x')] + (1 - D_j(x))$$

For any distribution $\phi$ over $B$, let $U(\phi, \cdot) = \mathbb{E}_{x \sim \phi}[U(x, \cdot)]$, and similarly for any distribution $\psi$ over $\{D_1, \ldots, D_N\}$, we will write $U(\cdot, \psi) = \mathbb{E}_{D \sim \psi}[U(\cdot, D)]$. Intuitively, the payoff function $U$ measures the predictive accuracy of the distinguisher in classifying whether the examples are drawn from the synthetic data player's distribution $\phi$ or the private dataset $X$. Thus, the synthetic data player aims to minimize $U$ while the distinguisher player aims to maximize $U$.

**Definition 2.** *The pair $(\overline{D}, \overline{\phi})$ is an $\alpha$-approximate equilibrium of the post-GAN game if*

$$\max_{D_j \in \mathcal{D}} U(\overline{\phi}, D_j) \leq U(\overline{\phi}, \overline{D}) + \alpha, \quad \text{and} \quad \min_{\phi \in \Delta(B)} U(\phi, \overline{D}) \geq U(\overline{\phi}, \overline{D}) - \alpha \quad (1)$$

By von Neumann's minimax theorem, there exists a value $V$ – called the *game value* – such that

$$V = \min_{\phi \in \Delta(B)} \max_{j \in [N]} U(\phi, D_j) = \max_{\psi \in \Delta(\mathcal{D})} \min_{x \in B} U(x, \psi)$$

The game value corresponds to the payoff value at an exact equilibrium of the game (that is $\alpha = 0$). When the set of discriminators cannot predict the real versus fake examples better than random guessing, the game value $V = 1$. We now show that under the assumption that the generated samples in $B$ approximately cover the support of the dataset $X$, the distinguisher player cannot distinguish the real and fake distributions much better than by random guessing.

**Theorem 1.** *Fix a private dataset $X \in (\mathbb{R}^d)^n$. Suppose that for every $x \in X$, there exists $x_b \in B$ such that $\|x - x_b\|_2 \leq \gamma$. Suppose $\mathcal{D}$ includes a discriminator network $D^{1/2}$ that outputs $1/2$ for all inputs, and assume that all networks in $\mathcal{D}$ are $L$-Lipschitz. Then there exists a distribution $\phi \in \Delta(B)$ such that $(\phi, D^{1/2})$ is a $L\gamma$-approximate equilibrium, and so $1 \leq V \leq 1 + L\gamma$.*

We defer the proof to the appendix. While the support coverage assumption is strong, we show empirically the synthetic data player's mixture distribution in an approximate equilibrium improves on the distribution given by the last generator $G_N$ even when the assumption does not hold. We now provide a method for computing an approximate equilibrium of the game.

### 3.2 BOOSTING VIA EQUILIBRIUM COMPUTATION.

Our post-GAN boosting (PGB) method computes an approximate equilibrium of the post-GAN zero-sum game by simulating the so-called *no-regret dynamics*. Over $T$ rounds the synthetic data

player maintains a sequence of distributions $\phi^1, \ldots, \phi^T$ over the set $B$, and the distinguisher plays a sequence of discriminators $D^1, \ldots, D^T$. At each round $t$, the distinguisher first selects a discriminator $D$ using the exponential mechanism $\mathcal{M}_E$ with the payoff $U(\phi^t, \cdot)$ as the score function. This will find an accurate discriminator $D^t$ against the current synthetic distribution $\phi^t$, so that the synthetic data player can improve the distribution. Then the synthetic data player updates its distribution to $\phi^t$ based on an online no-regret learning algorithm–the multiplicative weights (MW) method Kivinen & Warmuth (1997). We can view the set of generated examples in $B$ as a set of "experts", and the algorithm maintains a distribution over these experts and, over time, places more weight on the examples that can better "fool" the distinguisher player. To do so, MW updates the weight for each $x \in B$ with

$$\phi^{t+1}(x) \propto \phi^t \exp\left(-\eta U(x, D^t)\right) \propto \exp\left(\eta D^t(x)\right) \tag{2}$$

where $\eta$ is the learning rate. At the end, the algorithm outputs the average plays $(\overline{D}, \overline{\phi})$ for both players. We will show these form an approximate equilibrium of the post-GAN zero-sum game (Freund & Schapire, 1997).

---

**Algorithm 1** Differentially Private Post-GAN Boosting

---

**Require:** a private dataset $X \in \mathcal{X}^n$, a synthetic dataset $B$ generated by the set of generators $\mathcal{G}$, a collection of discriminators $\{D_1, \ldots, D_N\}$, number of iterations $T$, per-round privacy budget $\epsilon_0$, learning rate parameter $\eta$.

**Initialize** $\phi^1$ to be the uniform distribution over $B$

**for** $t = 1, \ldots, T$ **do**

   **Distinguisher player**: Run exponential mechanism $\mathcal{M}_E$ to select a discriminator $D^t$ using quality score $q(X, D_j) = U(\phi^t, D_j)$ and privacy parameter $\epsilon_0$.

   **Synthetic data player**: Multiplicative weights update on the distribution over $B$: for each example $b \in B$:
$$\phi^{t+1}(b) \propto \phi^t(b) \exp(\eta D^t(b))$$

Let $\overline{D}$ be the discriminator defined by the uniform average over the set $\{D^1, \ldots, D^T\}$, and $\overline{\phi}$ be the distribution defined by the average over the set $\{\phi^1, \ldots, \phi^T\}$

---

Note that the synthetic data player's MW update rule does not involve the private dataset, and hence is just a post-processing step of the selected discriminator $D^t$. Thus, the privacy guarantee follows from applying the advacned composition of $T$ runs of the exponential mechanism.[1]

**Theorem 2** (Privacy Guarantee). *For any $\delta \in (0, 1)$, the private MW post-amplification algorithm satisfies $(\epsilon, \delta)$-DP with $\epsilon = \sqrt{2\log(1/\delta)T}\epsilon_0 + T\epsilon_0(\exp(\epsilon_0) - 1)$.*

Note that if the private GAN training algorithm satisfies $(\epsilon_1, \delta_1)$-DP and the Private PGB method satisfies $(\epsilon_2, \delta_2)$-DP, then the entire procedure is $(\epsilon_1 + \epsilon_2, \delta_1 + \delta_2)$-DP.

We now show that the pair of average plays form an approximate equilibrium of the game.

**Theorem 3** (Approximate Equilibrium). *With probability $1 - \beta$, the pair $(\overline{D}, \overline{\phi})$ is an $\alpha$-approximate equilibrium of the post-GAN zero-sum game with $\alpha = 4\eta + \frac{\log|B|}{\eta T} + \frac{2\log(NT/\beta)}{n\epsilon_0}$. If $T \geq n^2$ and $\eta = \frac{1}{2}\sqrt{\log(|B|)/T}$, then*

$$\alpha = O\left(\frac{\log(nN|B|/\beta)}{n\epsilon_0}\right)$$

We provide a proof sketch here and defer the full proof to the appendix. By the result of Freund & Schapire (1997), if the two players have low regret in the dynamics, then their average plays form an approximate equilibrium, where the regret of the two players is defined as $R_{\mathrm{syn}} = \sum_{t=1}^{T} U(\phi^t, D^t) - \min_{b \in B} \sum_{t=1}^{T} U(b, D^t)$ and $R_{\mathrm{dis}} = \max_{D_j} \sum_{t=1}^{T} U(\phi^t, D_j) - \sum_{t=1}^{T} U(\phi^t, D^t)$. Then the approximate equilibrium guarantee directly follows from bounding $R_{\mathrm{syn}}$ with the regret bound of MW and $R_{\mathrm{dis}}$ with the approximate optimality of the exponential mechanism.

---

[1]Note that since the quality scores from the GAN Discriminators are assumed to be probabilities and the score function takes an average over $n$ probabilities (one for each private example), the sensitivity is $\Delta(q) = \frac{1}{n}$.

**Non-Private PGB.** The Private PGB method has a natural non-private variant: in each round, instead of drawing from the exponential mechanism, the distinguisher player will simply compute the exact best response: $D^t = \arg\max_{D_j} U(\phi^t, D_j)$. Then if we set learning rate $\eta = \frac{1}{2}\sqrt{\log(|B|)/T}$ and run for $T = \log(|B|)/\alpha^2$ rounds, the pair $(\overline{D}, \overline{\phi})$ returned is an $\alpha$-approximate equilibrium.

**Extension with Discriminator Rejection Sampling.** The mixture discriminator $\overline{D}$ at the equilibrium provides an accurate predictor on which samples are unlikely. As a result, we can use $\overline{D}$ to further improve the data distribution $\overline{\phi}$ by the *discriminator rejection sampling* (DRS) technique of Azadi et al. (2019). The DRS scheme in our setting generates a single example as follows: first draw an example $x$ from $\overline{\phi}$ (the proposal distribution), and then accept $x$ with probability proportional to $\overline{D}(x)/(1 - \overline{D}(x))$. Note that the optimal discriminator $D^*$ that distinguishes the distribution $\overline{\phi}$ from true data distribution $p_d$ will accept $x$ with probability proportional to $p_d(x)/p_{\overline{\phi}}(x) = D^*(x)/(1 - D^*(x))$. Our scheme aims to approximate this ideal rejection sampling by approximating $D^*$ with the equilibrium strategy $\overline{D}$, whereas prior work uses the last discriminator $D_N$ as an approximation.

## 4 EMPIRICAL EVALUATION

We empirically evaluate how both the Private and Non-Private PGB methods affect the utility of the generated synthetic data from GANs. We show two appealing advantages of our approach: 1) non-private PGB outperforms the last Generator of GANs, and 2) our approach can significantly improve the synthetic examples generated by a GAN under differential privacy.

**Datasets.** We assess our method with a toy dataset drawn from a mixture of 25 Gaussians, which is commonly used to evaluate the quality of GAN (Srivastava et al., 2017; Azadi et al., 2019; Turner et al., 2019) and synthesize MNIST images. We then turn to real datasets from the American Census, and a standard machine learning dataset (Titanic).

**Privacy budget.** For the tasks with privacy, we set the privacy budget to be the same across all algorithms. Since Private PGB requires additional privacy budget this means that the differentially private GAN training has to be stopped earlier as compared to running only a GAN to achieve the same privacy guarantee. Our principle is to allocate the majority of the privacy budget to the GAN training, and a much smaller budget for our Private PGB method. Throughout we used 80% to 90% of the final privacy budget on DP GAN training.[2]

**Utility measures.** Utility of synthetic data can be assessed along two dimensions; general utility and specific utility (Snoke et al., 2018; Arnold & Neunhoeffer, 2020). General utility describes the overall distributional similarity between the real data and synthetic datasets, but does not capture specific use cases of synthetic data. To assess general utility, we use the propensity score mean squared error (pMSE) measure (Snoke et al., 2018) (detailed in the Appendix). Specific utility of a synthetic dataset depends on the specific use an analyst has in mind. In general, specific utility can be defined as the similarity of results for analyses using synthetic data instead of real data. For each of the experiments we define specific utility measures that are sensible for the respective example. For the toy dataset of 25 gaussians we look at the number of high quality samples. For the American Census data we compare marginal distributions of the synthetic data to marginal distributions of the true data and look at the similarity of regression results.

### 4.1 EVALUATION OF NON-PRIVATE PGB

**Mixture of 25 Gaussians.** We first examine the performance of our approach on a two dimensional dataset with a mixture of 25 multivariate Gaussian distributions, each with a covariance matrix of $0.0025I$. The left column in Figure 1 displays the training data. Each of the 25 clusters consists

---

[2]Our observation is that the DP GAN training is doing the "heavy lifting". Providing a good "basis" for PGB requires a substantial privacy expenditure in training DP GAN. The privacy budget allocation is a hyperparameter for PGB that could be tuned. In general, the problem of differentially private hyperparameter selection is extremely important and the literature is thin (Liu & Talwar, 2019; Chaudhuri & Vinterbo, 2013).

of $1,000$ observations. The architecture of the GAN is the same across all results.[3] To compare the utility of the synthetic datasets with the real data, we inspect the visual quality of the resultsand calculate the proportion of high quality synthetic examples similar to Azadi et al. (2019),Turner et al. (2019) and Srivastava et al. (2017).[4]

Visual inspection of the results without privacy (in the top row of Figure 1) shows that our proposed method outperforms the synthetic examples generated by the last Generator of the GAN, as well as the last Generator enhanced with DRS. PGB over the last 100 stored Generators and Discriminators trained for $T = 1,000$ update steps, and the combination of PGB and DRS, visibly improves the results. The visual impression is confirmed by the proportion of high quality samples. The data from the last GAN generator have a proportion of $0.904$ high quality samples. The synthetic data after PGB achieves a higher score of $0.918$. The DRS samples have a proportion of $0.826$ high quality samples, and the combination of PGB and DRS a higher proportion of $0.874$ high quality samples.[5]

**MNIST Data.** We further evaluate the performance of our method on an image generation task with the MNIST dataset. Our results are based on the DCGAN GAN architecture (Radford et al., 2015) with the KL-WGAN loss (Song & Ermon, 2020). To evaluate the quality of the generated images we use a metric that is based on the Inception score (IS) (Salimans et al., 2016), where instead of the Inception Net we use a MNIST Classifier that achieves $99.65\%$ test accuracy. The theoretical best score of the MNIST IS is 10, and the real test images achieve a score of $9.93$. Without privacy the last GAN Generator achieves a score of $8.41$, using DRS on the last Generator slightly decreases the score to $8.21$, samples with PGB achieve a score of $8.76$, samples with the combination of PGB and DRS achieve a similar score of $8.77$ (all inception scores are calculated on 5,000 samples). Uncurated samples for all methods are included in the Appendix.

### 4.2 Evaluation of Private PGB

**Mixture of 25 Gaussians.** To show how the differentially private version of PGB improves the samples generated from GANs that were trained under differential privacy, we first re-run the experiment with the two-dimensional toy data.[6] Our final value of $\epsilon$ is 1 and $\delta$ is $\frac{1}{2N}$. For the results with PGB, the GAN training contributes $\epsilon_1 = 0.9$ to the overall $\epsilon$ and the Private PGB algorithm $\epsilon_2 = 0.1$. Again a first visual inspection of the results in Figure 1 (in the bottom row) shows that post-processing the results of the last GAN Generator with Private PGB is worthwhile. Private PGB over the last 100 stored Generators and Discriminators trained for $T = 1,000$ update steps, again, visibly improves the results. Again, our visual impression is confirmed by the proportion of high quality samples. The last Generator of the differentially private GAN achieves a proportion of $0.031$ high quality samples. With DRS on top of the last Generator, the samples achieve a quality score of $0.035$. The GAN enhanced with Private PGB achieves a proportion of $0.044$ high quality samples, the combination of Private PGB and DRS achieves a quality score of $0.053$.

**MNIST Data.** On the MNIST data, with differential privacy ($\epsilon = 10$, $\delta = \frac{1}{2N}$) the last DP GAN Generator achieves an inception score of $8.07$, using DRS on the last Generator the IS improves to $8.18$. With Private PGB the samples achieve an IS of $8.58$, samples with the combination of Private PGB and DRS achieve the highest IS of $8.66$.[7] Uncurated samples for all methods are included in the Appendix.

**Private Synthetic 1940 American Census Samples.** While the results on the toy dataset are encouraging, the ultimate goal of private synthetic data is to protect the privacy of actual persons in

---

[3] A description of the architecture is in the Appendix. The code for the GANs and the PGB algorithm will be made available on GitHub.

[4] Note that the scores in Azadi et al. (2019) and Turner et al. (2019) do not account for the synthetic data distribution across the 25 modes. We detail our evaluation of high quality examples in the Appendix.

[5] The lower scores for the DRS samples are due to the capping penalty in the quality metric. Without the capping penalty the scores are $0.906$ for the last Generator, $0.951$ for PGB , $0.946$ for DRS and $0.972$ for the combination of PGB and DRS.

[6] To achieve DP, we trained the Discriminator with a DP optimizer as implemented in `tensorflow_privacy` or the `opacus` library. We keep track of the values of $\epsilon$ and $\delta$ by using the moments accountant (Abadi et al., 2016; Mironov, 2017).

[7] All inception scores are calculated on $5,000$ samples.

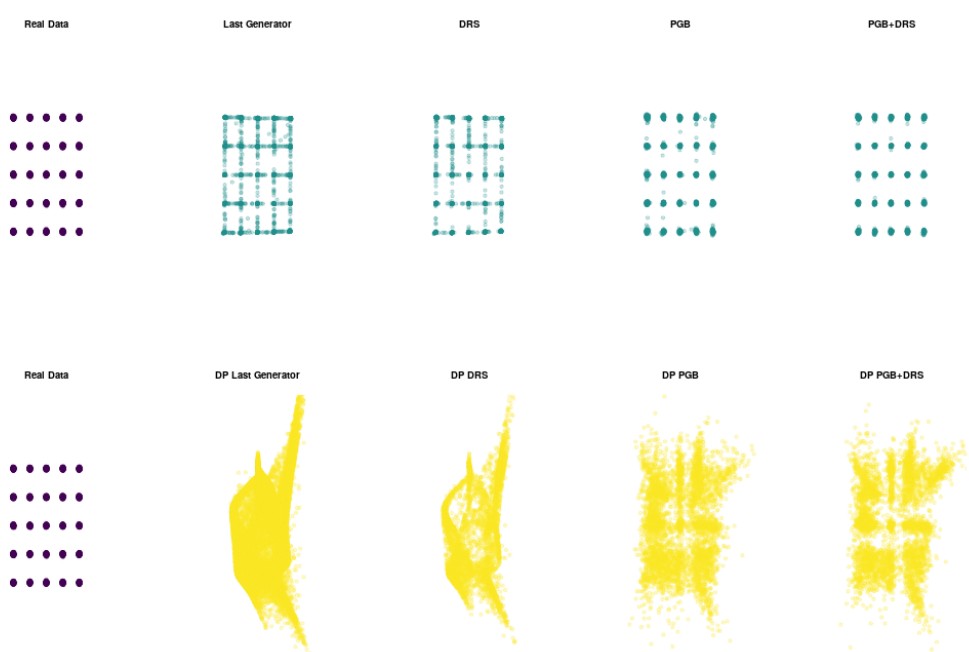

Figure 1: Real samples from 25 multivariate normal distributions, synthetic examples without privacy from a GAN, DRS, Non-Private PGB and the combination of PGB and DRS (top row). Synthetic examples from a GAN with differential privacy, DP DRS, Private PGB and the combination of Private PGB and DRS (bottom row).

data collections, and to provide useful data to interested analysts. In this section we report the results of synthesizing data from the 1940 American Census. We rely on the public use micro data samples (PUMS) as provided in Ruggles et al. (2019).[8] For 1940 we synthesize an excerpt of the 1% sample of all Californians that were at least 18 years old.[9] Our training sample consists of 39,660 observations and 8 attributes (sex, age, educational attainment, income, race, Hispanic origin, marital status and county). The test set contains another 9,915 observations. Our final value of $\epsilon$ is 1 and $\delta$ is $\frac{1}{2N} \approx 6.3 \times 10^{-6}$ (after DP GAN training with $\epsilon_1 = 0.8$ and PGB with $\epsilon_2 = 0.2, \delta_1 = \frac{1}{2N}, \delta_2 = 0$). The general utility scores as measured by the pMSE ratio score are $2.357$ (DP GAN), $2.313$ (DP DRS), $2.253$ (DP PGB), and $2.445$ (DP PGB+DRS). This indicates that PGB achieves the best general utility. To assess the specific utility of our synthetic census samples we compare one-way marginal distributions to the same marginal distributions in the original data. In panel (A) of Figure 2 we show the distribution of race membership. Comparing the synthetic data distributions to the true distribution (in gray), we conclude that PGB, improves upon the last Generator. To underpin the visual impression we calculate the total variation distance between each of the synthetic distributions and the real distribution, the data from the last GAN Generator has a total variation distance of $0.58$, DP DRS of $0.44$, DP PGB of $0.22$ and DP PGB+DRS of $0.13$. Furthermore, we evaluate whether more complex analysis models, such as regression models, trained on synthetic samples could be used to make sensible out-of-sample predictions. Panel (B) of Figure 2 shows a parallel coordinate plot to compare the out-of-sample root mean squared error of regression models trained on real data and trained on synthetic data. The lines show the RMSE for predicted income for all linear regression models trained with three independent variables from the set of on the synthetic data generated with Private PGB as compared to the last GAN generator and other post processing methods like DRS.

---

[8]Further experiments using data from the 2010 American Census can be found in the appendix.

[9]A 1% sample means that the micro data contains 1% of the total American (here Californian) population.

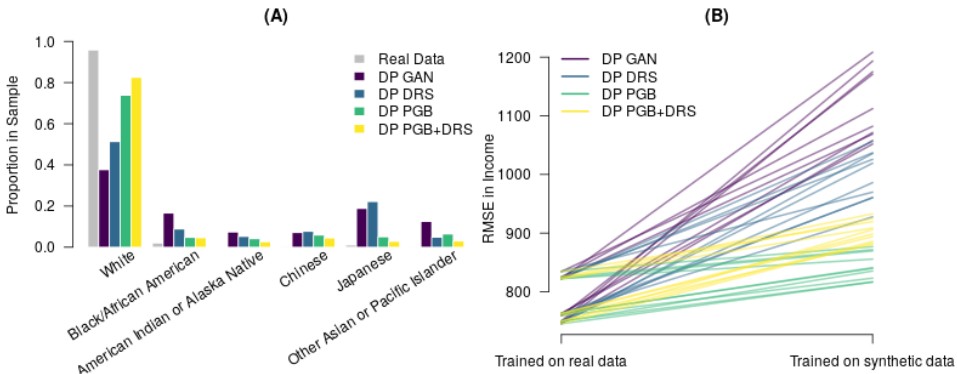

Figure 2: Specific Utility of Synthetic 1940 American Census Data. Panel (A): Distribution of Race Membership in Synthetic Samples. Panel (B): Regression RMSE with Synthetic Samples.

**Machine Learning Prediction with Synthetic Data.** In a final set of experiments we evaluate the performance of machine learning models trained on synthetic data (with and without privacy) and tested on real out-of-sample data. We synthesize the Kaggle Titanic[10] training set (891 observations of Titanic passengers on 8 attributes) and train three machine learning models (Logistic Regression, Random Forests (RF) (Breiman, 2001) and XGBoost (Chen & Guestrin, 2016)) on the synthetic datasets to predict whether someone survived the Titanic catastrophe. We then evaluate the performance on the test set with 418 observations. To address missing values in both the training set and the test set we independently impute values using the MissForest (Stekhoven & Bühlmann, 2012) algorithm. For the private synthetic data our final value of $\epsilon$ is 2 and $\delta$ is $\frac{1}{2N}$ (for PGB this implies DP GAN training with $\epsilon_1 = 1.6$ and PGB $\epsilon_2 = 0.4$). The models trained on synthetic data generated with our approaches (PGB and PGB+DRS) consistently perform better than models trained on synthetic data from the last generator or DRS – with or without privacy.[11]

ACKNOWLEDGMENTS

This work began when the authors were at the Simons Institute participating in the "Data Privacy: Foundations and Applications" program. We thank Thomas Steinke, Adam Smith, Salil Vadhan, and the participants of the DP Tools meeting at Harvard for helpful comments. Marcel Neunhoeffer is supported by the University of Mannheim's Graduate School of Economic and Social Sciences funded by the German Research Foundation. Zhiwei Steven Wu is supported in part by an NSF S&CC grant 1952085, a Google Faculty Research Award, and a Mozilla research grant. Cynthia Dwork is supported by the Alfred P. Sloan Foundation, "Towards Practicing Privacy" and NSF CCF-1763665.

---

[10]https://www.kaggle.com/c/titanic/data

[11]Table 1 in the appendix summarizes the results in more detail. We present the accuracy, ROC AUC and PR AUC to evaluate the performance.

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

# A  PROOFS

## A.1  PROOF OF THEOREM 1

*Proof of Theorem 1.* Note that if the synthetic data player plays the distribution over $X$, then $U(p_X, D) = \mathbb{E}_{x \sim p_X}[D(x')] + \mathbb{E}_{x \sim \phi}[1 - D(x)] = 1$ for any discriminator $D \in \mathcal{D}$. Now let us replace each element in $X$ with its $\gamma$-approximation in $B$ and obtain a new dataset $X_B$, and let $p_{X_B}$ denote the empirical distribution over $X_B$. By the Lipschitz conditions, we then have $|U(p_X, D) - U(p_{X_B}, D)| \leq L\gamma$. This means $U(p_{X_B}, D) \in [1 - L\gamma, 1 + L\gamma]$ for all $D$. Also, for all $\phi \in \Delta(B)$, we have $U(\phi, D^{1/2}) = 1$. Thus, $(p_{X_b}, D^{1/2})$ satisfies equation 1 with $\alpha = L\gamma$. $\qquad\square$

## A.2  PROOF OF THE APPROXIMATE EQUILIBRIUM

*Proof.* We will use the seminal result of Freund & Schapire (1997), which shows that if the two players have low regret in the dynamics, then their average plays form an approximate equilibrium. First, we will bound the regret from the data player. The regret guarantee of the multiplicative weights algorithm (see e.g. Theorem 2.3 of Arora et al. (2012)) gives

$$\sum_{t=1}^{T} U(\phi^t, D^t) - \min_{b \in B} \sum_{t=1}^{T} U(b, D^t) \leq 4\eta T + \frac{\log |B|}{\eta} \tag{3}$$

Next, we bound the regret of the distinguisher using the accuracy guarantee of the exponential mechanism (McSherry & Talwar, 2007). For each $t$, we know with probability $(1 - \beta/T)$,

$$\max_{D_j} U(\phi^t, D_j) - U(\phi^t, D^t) \leq \frac{2 \log(NT/\beta)}{n\epsilon_0}$$

Taking a union bound, we have this accuracy guarantee holds for all $t$, and so

$$\max_{D_j} \sum_{t=1}^{T} U(\phi^t, D_j) - \sum_{t=1}^{T} U(\phi^t, D^t) \leq \frac{2T \log(NT/\beta)}{n\epsilon_0} \tag{4}$$

Then following the result of Freund & Schapire (1997), their average plays $(\overline{D}, \overline{\phi})$ is an $\alpha$-approximate equilibrium with

$$\alpha = 4\eta + \frac{\log |B|}{\eta T} + \frac{2 \log(NT/\beta)}{n\epsilon_0}$$

Plugging in the choices of $T$ and $\eta$ gives the stated bound. $\qquad\square$

# B  ADDITIONAL DETAILS ON THE QUALITY EVALUATION

## B.1  ON THE CALCULATION OF THE PMSE.

To calculate the pMSE one trains a discriminator to distinguish between real and synthetic examples. The predicted probability of being classified as real or synthetic is the propensity score. Taking all propensity scores into account the mean squared error between the propensity scores and the proportion of real data examples is calculated. A synthetic dataset has high general utility, if the model can at best predict probabilities of 0.5 for both real and synthetic examples, then the pMSE would be 0.

## B.2  SPECIFIC UTILITY MEASURE FOR THE 25 GAUSSIANS.

In the real data, given the data generating process outlined in section 4.1, at each of the 25 modes 99% of the observations lie within a circle with radius $r = \sqrt{0.0025 \cdot 9.21034}$ around the mode centroids, where 9.21034 is the critical value at $p = 0.99$ of a $\chi^2$ distribution with 2 degrees of freedom, and 0.0025 is the variance of the spherical gaussian.

To calculate the quality score we count the number of observations within each of these 25 circles. If one of the modes contains more points than we would expect given the true distribution the count is capped accordingly. Our quality score for the toy dataset of 25 gaussians can be expressed as $Q = \sum_i^{25}(\min(p_{real}^i \cdot N_{syn}, N_{syn}^i)/N_{syn})$, where $i$ indexes the clusters, $p_{real}$ is the true distribution of points per cluster, $N_{syn}^i$ the number of observations at a cluster within radius $r$, and $N_{syn}$ the total number of synthetic examples.

## C  GAN ARCHITECTURES

### C.1  DETAILS ON THE EXPERIMENTS WITH THE 25 GAUSSIANS.

The generator and discriminator are neural nets with two fully connected hidden layers (Discriminator: 128, 256; Generator: 512, 256) with Leaky ReLu activations. The latent noise vector Z is of dimension 2 and independently sampled from a gaussian distribution with mean 0 and standard deviation of 1. For GAN training we use the KL-WGAN loss (Song & Ermon, 2020). Before passing the Discriminator scores to PGB we transform them to probabilities using a sigmoid activation.

### C.2  GAN ARCHITECTURE FOR THE 1940 AMERICAN CENSUS DATA.

The GAN networks consist of two fully connected hidden layers (256, 128) with Leaky ReLu activation functions. To sample from categorical attributes we apply the Gumbel-Softmax trick (Maddison et al., 2016; Jang et al., 2016) to the output layer of the Generator. We run our PGB algorithm over the last 150 stored Generators and Discriminators and train it for $T = 400$ update steps.

## D  PRIVATE SYNTHETIC 2010 AMERICAN DECENNIAL CENSUS SAMPLES.

We conducted further experiments on more recent Census files. The 2010 data is similar to the data that the American Census is collecting for the 2020 decennial Census. For this experiment, we synthesize a 10% sample for California with 3,723,669 observations of 5 attributes (gender, age, Hispanic origin, race and puma district membership). Our final value of $\epsilon$ is 0.795 and $\delta$ is $\frac{1}{2N} \approx 1.34 \times 10^{-7}$ (for PGB the GAN training contributes $\epsilon = 0.786$ and PGB $\epsilon = 0.09$). The pMSE ratio scores are 1.934 (DP GAN), 1.889 (DP DRS), 1.609 (DP PGB) and 1.485 (DP PGB+DRS), here PGB achieves the best general utility. For specific utility, we compare the accuracy of three-way marginals on the synthetic data to the proportions in the true data.[12] We tabulate race (11 answer categories in the 2010 Census) by Hispanic origin (25 answer categories in the 2010 Census) by gender (2 answer categories in the 2010 Census) giving us a total of 550 cells. To assess the specific utility for these three-way marginals we calculate the average accuracy across all 550 cells. Compared to the true data DP GAN achieves 99.82%, DP DRS 99.89%, DP PGB 99.89% and the combination of DP PGB and DRS 99.93%. Besides the average accuracy across all 550 cells another interesting metric of specific utility is the number of cells in which each synthesizer achieves the highest accuracy compared to the other methods, this is the case 43 times for DP GAN, 30 times for DP DRS, 90 times for DP PGB and 387 times for DP PGB+DRS. Again, this shows that our proposed approach can improve the utility of private synthetic data.

## E  DETAILED RESULTS OF MACHINE LEARNING PREDICTION WITH SYNTHETIC DATA

Table 1 summarizes the results for the machine learning prediction experiment with the Titanic data. We present the accuracy, ROC AUC and PR AUC to evaluate the performance. It can be seen that the models trained on synthetic data generated with our approach consistently perform better than models trained on synthetic data from the last generator or DRS – with or without privacy. To put these values into perspective, the models trained on the real training data and tested on the same out-of-sample data achieve the scores in table 2.

---

[12]A task that is similar to the tables released by the Census.

Table 1: Predicting Titanic Survivors with Machine Learning Models trained on synthetic data and tested on real out-of-sample data. Median scores of 20 repetitions with independently generated synthetic data. With differential privacy $\epsilon$ is 2 and $\delta$ is $\frac{1}{2N} \approx 5.6 \times 10^{-4}$.

| | GAN | DRS | PGB | PGB + DRS |
|---|---|---|---|---|
| Logit Accuracy | 0.626 | 0.746 | 0.701 | **0.765** |
| Logit ROC AUC | 0.591 | 0.760 | 0.726 | **0.792** |
| Logit PR AUC | 0.483 | 0.686 | 0.655 | **0.748** |
| RF Accuracy | 0.594 | 0.724 | 0.719 | **0.742** |
| RF ROC AUC | 0.531 | 0.744 | 0.741 | **0.771** |
| RF PR AUC | 0.425 | 0.701 | 0.706 | **0.743** |
| XGBoost Accuracy | 0.547 | 0.724 | 0.683 | **0.740** |
| XGBoost ROC AUC | 0.503 | 0.732 | 0.681 | **0.772** |
| XGBoost PR AUC | 0.400 | 0.689 | 0.611 | **0.732** |
| | DP GAN | DP DRS | DP PGB | DP PGB +DRS |
| Logit Accuracy | 0.566 | 0.577 | 0.640 | **0.649** |
| Logit ROC AUC | 0.477 | 0.568 | 0.621 | **0.624** |
| Logit PR AUC | 0.407 | 0.482 | 0.532 | **0.547** |
| RF Accuracy | 0.487 | 0.459 | 0.481 | **0.628** |
| RF ROC AUC ROC AUC | 0.512 | 0.553 | 0.558 | **0.652** |
| RF PR AUC PR AUC | 0.407 | 0.442 | 0.425 | **0.535** |
| XGBoost Accuracy | 0.577 | 0.589 | 0.609 | **0.641** |
| XGBoost ROC AUC | 0.530 | 0.586 | **0.619** | 0.596 |
| XGBoost PR AUC | 0.398 | 0.479 | 0.488 | **0.526** |

Table 2: Predicting Titanic Survivors with Machine Learning Models trained on real data and tested on real out-of-sample data.

| Model | Score |
|---|---|
| Logit Accuracy | 0.764 |
| Logit ROC AUC | 0.813 |
| Logit PR AUC | 0.785 |
| RF Accuracy | 0.768 |
| RF ROC AUC | 0.809 |
| RF PR AUC | 0.767 |
| XGBoost Accuracy | 0.768 |
| XGBoost ROC AUC | 0.773 |
| XGBoost PR AUC | 0.718 |

# F SYNTHETIC MNIST SAMPLES

Figure 3 shows uncurated samples from the last Generator after 30 epochs of training without differential privacy in Panel 3a and with differential privacy ($\epsilon =$, $\delta =$) in Panel 3b. Figure 4 shows uncurated samples with DRS on the last Generator. Figure 5 shows uncurated samples after PGB and Figure 6 shows uncurated samples after the combination of PGB and DRS. In Figure 7 we show the 100 samples with the highest PGB probabilities.

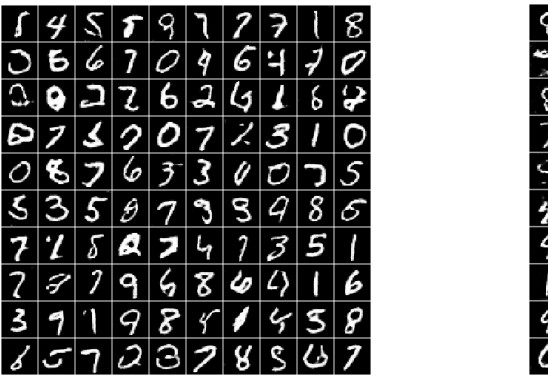

(a) Without Differential Privacy.    (b) With Differential Privacy ($\epsilon = 10, \delta = \frac{1}{2N}$)

Figure 3: Uncurated MNIST samples from last GAN Generator after 30 epochs.

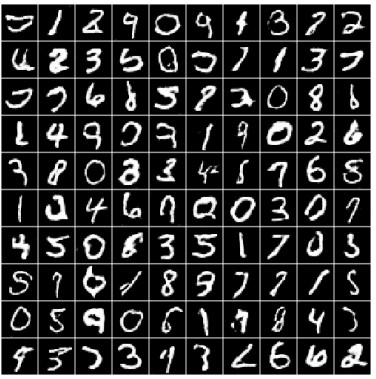

(a) Without Differential Privacy.    (b) With Differential Privacy ($\epsilon = 10, \delta = \frac{1}{2N}$)

Figure 4: Uncurated MNIST samples with DRS on last Generator after 30 epochs.

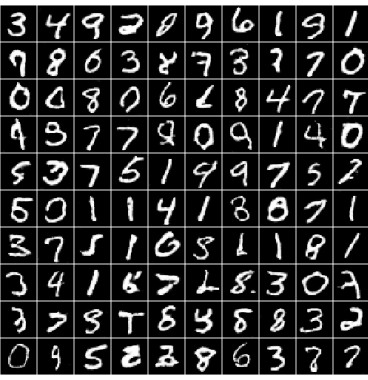

(a) Without Differential Privacy.    (b) With Differential Privacy ($\epsilon = 10, \delta = \frac{1}{2N}$)

Figure 5: Uncurated MNIST samples with PGB after 30 epochs (without DP) and 25 epochs (with DP).

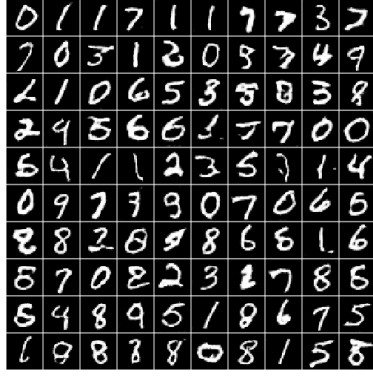

(a) Without Differential Privacy.

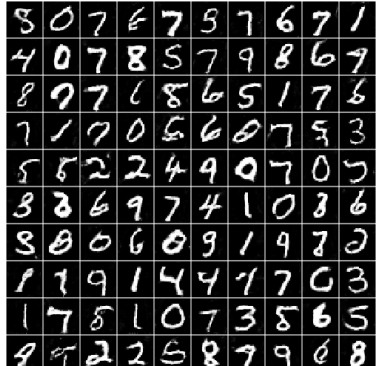

(b) With Differential Privacy ($\epsilon = 10$, $\delta = \frac{1}{2N}$)

Figure 6: Uncurated MNIST samples with PGB and DRS after 30 epochs (without DP) and 25 epochs (with DP).

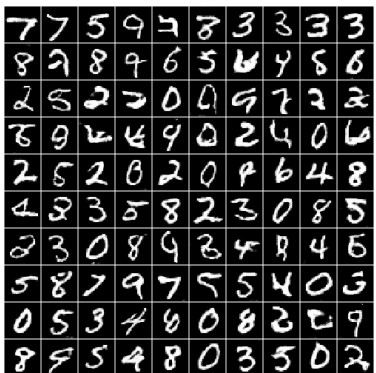

(a) Without Differential Privacy.

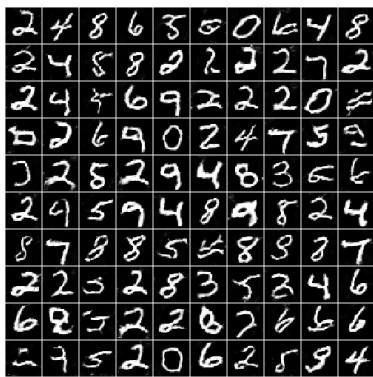

(b) With Differential Privacy ($\epsilon = 10$, $\delta = \frac{1}{2N}$)

Figure 7: Top 100 MNIST samples after PGB after 30 epochs (without DP) and 25 epochs (with DP).

