# OpenReview forum: "Private Post-GAN Boosting"
_ICLR.cc/2021/Conference — ICLR 2021 Poster_

### Official Review · AnonReviewer2 · 2020-10-27
**A qualitative comparison with private generative models on MNIST is needed**

**Rating:** 6
**Confidence:** 2

**Review:**

Summary: This paper studies the differential private synthetic dataset generation. Unlike previous DP based GAN models, this paper aims to boost the sample quality of after the training stage. In particular, the final synthetic dataset is sampled from the sequence of generators obtained during GAN training. The distribution is obtained by a private two-player game between the privately selected discriminator and a sampler from the mixture of generators. The results are demonstrated on gaussian data and tabular data.


Pros: 1. The sample quality of private generative models is known to be not as good as non-private models. This paper provides a practical private post gan boosting algorithm to improve the sample quality.

Cons:1. My main concern is on experiments. It is known that private generative models have bad sample quality on image data. Prior works on private synthetic generation papers usually show results on MNIST. It would be better if the authors could compare private PGB on MNIST dataset.

2. It would be better to have an ablation study of the proposed PGB and discriminator rejection sampling. For example, in Figure~1, the baselines for both non-private gan and private gan are too bad. I am wondering whether the gain is from rejection sampling or the proposed PGB algorithm.


Questions:

1. I am curious about how to split epsilon for gan training and the post gan boosting. Are there any principled reasons for the split?

2. How do you calculate the sensitivity for the exponential mechanism?

---

> ### Author Response · Authors · 2020-11-24
> **Response to Reviewer 2**
>
> Thank you very much for your helpful comments. We uploaded an updated version of the manuscript. We hope this addresses your concerns.
>
> *1. My main concern is on experiments. It is known that private generative models have bad sample quality on image data. Prior works on private synthetic generation papers usually show results on MNIST. It would be better if the authors could compare private PGB on MNIST dataset.*
>
> In the updated manuscript we included an evaluation of Private PGB on the MNIST data. Our results show that our proposed Private PGB method improves the quality of synthetic samples. We included these results on page 8 of the updated manuscript. Uncurated samples of DP MNIST digits can be found in the Appendix.
>
> *2. It would be better to have an ablation study of the proposed PGB and discriminator rejection sampling. For example, in Figure~1, the baselines for both non-private gan and private gan are too bad. I am wondering whether the gain is from rejection sampling or the proposed PGB algorithm.*
>
> We included an ablation study to directly compare the samples from the last Generator, DRS, PGB and the combination of PGB and DRS on both the toy dataset of 25 gaussians, as well as on the MNIST data (without and with differential privacy). The results show that the improvements mainly come from our proposed PGB method. We included these results in Figure 1 on page 7 of the updated manuscript and in the respective sections describing the results of the toy data experiments and the MNIST experiments.
>
> *Q: I am curious about how to split epsilon for gan training and the post gan boosting. Are there any principled reasons for the split?*
>
> Our principle is to allocate the majority of the privacy budget to the DP GAN training, and a much smaller budget for our PGB method. Our intuition is that the DP GAN training is doing the "heavy lifting": providing a good "basis" for PGB requires a substantial privacy expenditure in training DP GAN. Without a good basis, PGB cannot provide good quality synthetic data. We address this in Footnote 2 on page 6 of the manuscript. Assessing the impact of the privacy budget allocation between DP GAN training and PGB is definitely worthwhile and something we want to address in future work.
>
> *Q: How do you calculate the sensitivity for the exponential mechanism?*
>
> Regarding the sensitivity for the exponential mechanism, as stated in Section 2.1 on page 3, for Private PGB we assume the discriminator scores to be probabilities.  Since the score function takes an average over n probabilities (one for each private example), the sensitivity is 1/n. We added footnote 1 in Section 3.2 (page 5) to highlight this.

---

### Official Review · AnonReviewer1 · 2020-10-29
**Good progress in differentially private GAN**

**Rating:** 7
**Confidence:** 3

**Review:**

This paper proposes an algorithm to process the sequence of generators and discriminators produced by any differentially private GAN training, in order to produce synthetic data of better quality. The paper also presents some empirical evaluations.
There have been a few works on differentially private GAN, yet the noised required by differential privacy usually causes significant degradation in utility. So an algorithm for improving the results can be quite valuable. The evaluation was mainly on simple datasets, which is understandable as differentially private GAN can be quite hard. However, I think it might be helpful to do more experiments on more difficult datasets (like those usually used to evaluate non-private GAN) with various epsilons (even with very large epsilon), so that readers and future researchers can understand the limitations of the current sota.

The presentation is clear. The privacy aspect of the algorithm, e.g. which part needs privacy protection and what is the sensitivity of the score function, can be elaborated more.

---

> ### Author Response · Authors · 2020-11-24
> **Response to Reviewer 1**
>
> Thank you very much for your positive and helpful review. We updated the manuscript and hope that the updated version addresses the points you raised.
>
> *Q: However, I think it might be helpful to do more experiments on more difficult datasets (like those usually used to evaluate non-private GAN) with various epsilons (even with very large epsilon), so that readers and future researchers can understand the limitations of the current sota.*
>
> In the updated manuscript we included an evaluation of Private PGB on the MNIST data. Our results show that our proposed Private PGB method improves the quality of synthetic samples. We included these results on page 8 of the updated manuscript. Uncurated samples of DP MNIST digits can be found in the Appendix.
>
> *Q: The privacy aspect of the algorithm, e.g. which part needs privacy protection and what is the sensitivity of the score function, can be elaborated more.*
>
> As stated in Section 2.1 on page 3, for Private PGB we assume the discriminator scores to be probabilities.  Since the score function takes an average over n probabilities (one for each private example), the sensitivity is 1/n. We added footnote 1 in Section 3.2 (page 5) to highlight this.

---

### Official Review · AnonReviewer3 · 2020-11-02
**Sound theory, good results**

**Rating:** 8
**Confidence:** 4

**Review:**

The paper proposes a method of improving the generated samples of differential-private synthetic dataset using GANs by boosting them post training. They support their proposed method using theory, and then empirically show that it works on 3 types of machine learning tasks.

This paper presents a novel way of utilizing the sequence of generators and discriminators during training as they are already part of the privacy budget. So it significantly improves the quality of GAN-generated samples for different experiments under the same privacy budget.

The experiments provide evidence of the utility of the proposed method in all three tasks. The community can definitely benefit from this paper.

---

> ### Author Response · Authors · 2020-11-24
> **Response to Reviewer 3**
>
> Thank you very much for your positive review.

---

### Author Response · Authors · 2020-11-24
**Reply to Reviewers | Private Post-GAN Boosting**

We thank the three reviewers for their positive and helpful reviews.

In the following we will address the points raised by Reviewers 1 & 2.

We agree with reviewers 1 & 2 that an evaluation of Private PGB on an image data set could highlight the advantage of our proposed method. In the updated manuscript we included an evaluation of Private PGB on the MNIST data as suggested by Reviewer 2 (and implied by Reviewer 1). Our results show that our proposed Private PGB method improves the quality of synthetic samples. We included these results on page 8 of the updated manuscript. Uncurated samples of DP MNIST digits can be found in the Appendix.

Yet, ultimately, our goal with differentially private synthetic data is mainly to generate tabular data to facilitate research with privacy guarantees. Working with noisy statistics released under differential privacy requires training. With differentially private synthetic data researchers could potentially keep their established workflows, while still protecting privacy. The focus of our manuscript was, therefore, on applied downstream tasks as shown in the experiments with the American Census data and the Titanic data.

Addressing the second point of Reviewer 2, we included an ablation study to directly compare the samples from the last Generator, DRS, PGB and the combination of PGB and DRS on both the toy dataset of 25 gaussians, as well as on the MNIST data (without and with differential privacy). The results show that the improvements mainly come from our proposed PGB method. We included these results in Figure 1 on page 7 of the updated manuscript and in the respective sections describing the results of the toy data experiments and the MNIST experiments.

*Q:  How do you calculate the sensitivity for the exponential mechanism?/ The privacy aspect of the algorithm, e.g. which part needs privacy protection and what is the sensitivity of the score function, can be elaborated more.*

A: Regarding the sensitivity of the score function/exponential mechanism, as stated in Section 2.1 on page 3, for Private PGB we assume the discriminator scores to be probabilities. Since the score function takes an average over n probabilities (one for each private example), the sensitivity is 1/n. We added footnote 1 in Section 3.2 (page 5) to highlight this.

*Q: I am curious about how to split epsilon for gan training and the post gan boosting. Are there any principled reasons for the split?*

A: Our principle is to allocate the majority of the privacy budget to the DP GAN training, and a much smaller budget for our PGB method. Our intuition is that the DP GAN training is doing the “heavy lifting”: providing a good “basis” for PGB requires a substantial privacy expenditure in training DP GAN. Without a good basis, PGB cannot provide good quality synthetic data. We address this in Footnote 2 on page 6 of the manuscript.
Assessing the impact of the privacy budget allocation between DP GAN training and PGB is definitely worthwhile and something we want to address in future work.

---

### Decision · Program_Chairs · 2021-01-07
**Final Decision**

**Decision:**

Accept (Poster)

**Comment:**

This paper provides a privacy-preserving method to boost the sample quality after training a GAN. The reviewers were unanimous that this paper should be presented at ICLR, with an important contribution to privacy-preserving GANs.